# A computational study on the role of glutamate and NMDA receptors on cortical spreading depression using a multidomain electrodiffusion model

**Austin Tuttle** [1], **Jorge Riera Diaz** [2], **Yoichiro Mori** [1,3,4]*

**1** School of Mathematics, University of Minnesota, Minneapolis, Minnesota, United States of America,
**2** Department of Biomedical Engineering, Florida International University, Miami, Florida, United States of America, **3** Department of Mathematics, University of Pennsylvania, Philadelphia, Pennsylvania, United States of America, **4** Department of Biology, University of Pennsylvania, Philadelphia, Pennsylvania, United States of America

* ymori@umn.edu

**Data Availability Statement:** All data in the paper is available at: https://upenn.box.com/s/22pjg7a85lw1df0szb9nbyfcrpfmq6mw. Code for the simulations is available in:

## Abstract

Cortical spreading depression (SD) is a spreading disruption of ionic homeostasis in the brain during which neurons experience complete and prolonged depolarizations. SD is the basis of migraine aura and is increasingly associated with many other brain pathologies. Here, we study the role of glutamate and NMDA receptor dynamics in the context of an ionic electrodiffusion model. We perform simulations in one (1D) and two (2D) spatial dimension. Our 1D simulations reproduce the "inverted saddle" shape of the extracellular voltage signal for the first time. Our simulations suggest that SD propagation depends on two overlapping mechanisms; one dependent on extracellular glutamate diffusion and NMDA receptors and the other dependent on extracellular potassium diffusion and persistent sodium channel conductance. In 2D simulations, we study the dynamics of spiral waves. We study the properties of the spiral waves in relation to the planar 1D wave, and also compute the energy expenditure associated with the recurrent SD spirals.

## Author summary

Cortical spreading depression is a wave of neuronal silencing and ion concentration changes that sweeps slowly through the brain. It is the basis of migraine aura, in which a migraine patient sees a defect move through ones visual field 30 minutes to an hour prior to the headache attack. In this paper, we study the mechanisms by which cortical spreading depression travels as a wave through the brain. We construct a detailed mathematical model based on the physics of ion movement as well as what is known about the molecular players of cortical spreading depression. We build a simulation program that successfully solves the resulting set of highly coupled equations. We find that cortical spreading depression can propagate as a wave by distinct but overlapping mechanisms. We also simulate spiral cortical spreading depression waves and study their properties.

https://github.com/ADTuttle/2d_CSD. Further details about the code can be found in S3 Text.

**Funding:** A.T. and Y.M. were supported by the National Science Foundation, grant number DMS 1516978 and J.R.D. was supported by the National Science Foundation grant number DMS 1516176. The National Science Foundation, DMS (Division of Mathematical Sciences) website is: https://www.nsf.gov/div/index.jsp?div=DMS. The funders had no role in study design, data collection and analysis, decision to publish, or preparation of the manuscript.

**Competing interests:** The authors have declared that no competing interests exist.

## Introduction

Cortical Spreading Depression (SD) is a pathophysiological phenomenon in the central nervous system characterized by a local breakdown in ionic homeostasis resulting in a temporary silencing of neuronal electrical activity. This local ionic disruption propagates at speeds of 2-7 mm/min [1]. SD is the physiological substrate of migraine aura [2, 3]. SD and related phenomena have also been linked to other disease conditions in the brain including traumatic brain injury, ischemic stroke, subarachnoid hemorrhage, and intracerebral hematoma [1, 4]. SD was first discovered by Leao in 1944 [5], and has since been intensively studied from both experimental and theoretical points of view [6]. Despite this long history, many aspects of SD still remain elusive [6–10].

Classical models of Hodgkin-Huxley type, suitable for the description of normal electrophysiological activity, cannot be used for SD, which has a much slower time scale and features very large ionic concentration deflections. Many past theoretical models of SD are of reaction-diffusion type [4, 11, 12]; such models in essence treat ions as being uncharged, and therefore cannot capture some important aspects of SD propagation. To address these difficulties, we developed in previous papers an electrodiffusive model of SD [13, 14]. The electrodiffusive description of ionic balance allows us to better capture the relevant biophysics, and we have thereby been successful in computing the extracellular voltage shift [13]. In [14], we introduced a model in which three compartments, the neuronal, glial, and extracellular compartments were considered. In this paper, we introduce two major extensions to our previous model; we add glutamate dynamics and extend our numerical method to handle simulations in two space dimensions.

Glutamate has long been suggested to play an important role in SD. Glutamate was one of the earliest suggested agents for SD initiation and propagation [15]. It has also become well-established that NMDA receptors (NMDAR) play an important role in SD. Indeed, the review article [6] asserts that, at least in normoxic tissue, NMDAR activation serves as the crucial link that allows SD initiation and propagation. On the other hand, most computational models of SD do not include glutamate dynamics (but see [16]), and have often relied on the activation of persistent Na (NaP) current for SD initiation and propagation [17, 18]. Here, we incorporate glutamate dynamics into our model, and examine the relative importance of NaP and NMDAR currents in SD activation.

Our computational results suggest that the there are two mechanisms that allow SD propagation. One relies on NaP activation and extracellular potassium (K) diffusion and the other relies on NMDAR activation and glutamate diffusion. These two mechanisms, however, cannot be cleanly separated. Even in the absence of NaP currents, in which case SD activation is solely due to NMDAR activation, K diffusion does play a role. When NaP and NMDAR currents coexist, these two mechanisms operate in parallel, and in this sense, our results can be seen as supporting Van Harreveld's dual hypothesis [19]. Indeed, we have found that the "inverted saddle" signature of the extracellular voltage shift often seen in SD measurements can be explained by the coexistence of these two mechanisms. The first valley corresponds primarily to NaP activation and the latter valley to NMDAR activation.

The above study on the NaP and NMDAR currents were conducted using 1D simulations. We further extend our model to 2D. While several detailed models have investigated recurrent SD in either 1D or 0D [16, 20], none have investigated recurrence in 2D (however, there have been phenomenological models [4]). Recurrence in 2D is significantly more complex than 1D, as the recurrence can arise from the geometry of the wave (spirals). These spirals can form around anatomical blocks (a physical boundary) or functional blocks (e.g previously ischemic regions). These recurrent waves are of great clinical importance as these repeated

depolarizations can exacerbate the damage caused by stroke or traumatic injuries [21–23]. We investigate the behavior of these spirals and highlight the differences to the 1D case. Furthermore, we study the energetics of SD [24], which is made possible by the fact that our biophysical model carries a natural thermodynamic structure [13]. We investigate the work done by ion pumps during SD, which correlates to the amount of stress repeated depolarization place on brain tissue.

Our model constitutes a nonlinear and highly coupled partial differential algebraic system, which we call the *multidomain electrodiffusion model* [13]. From a technical standpoint, our main contribution is the successful development and implementation of an efficient algorithm for this model. In contrast to our previous work [13, 14], we replace the algebraic constraint by its time derivative allowing for more stable and efficient computation. For the requisite linear algebra routines, we use PETSc [25], which provides a powerful suite of Krylov subspace solvers and their attendant preconditioners.

## Electrodiffusion model

The model we use here is based on work in [13, 14]. Let $\Omega$ be our domain in $\mathbb{R}^2$. We treat brain tissue as a multi-phasic continuum with three interpenetrating compartments, the neuronal ($k = 1$ or $k = n$), glial ($k = 2$ or $k = g$) and extracellular spaces ($k = 3$ or $k = e$) (see Fig 1). At each point in space we assign a volume fraction $\alpha_k$ and impose:

$$\sum_{k=1}^{3} \alpha_k(x, t) = 1. \tag{1}$$

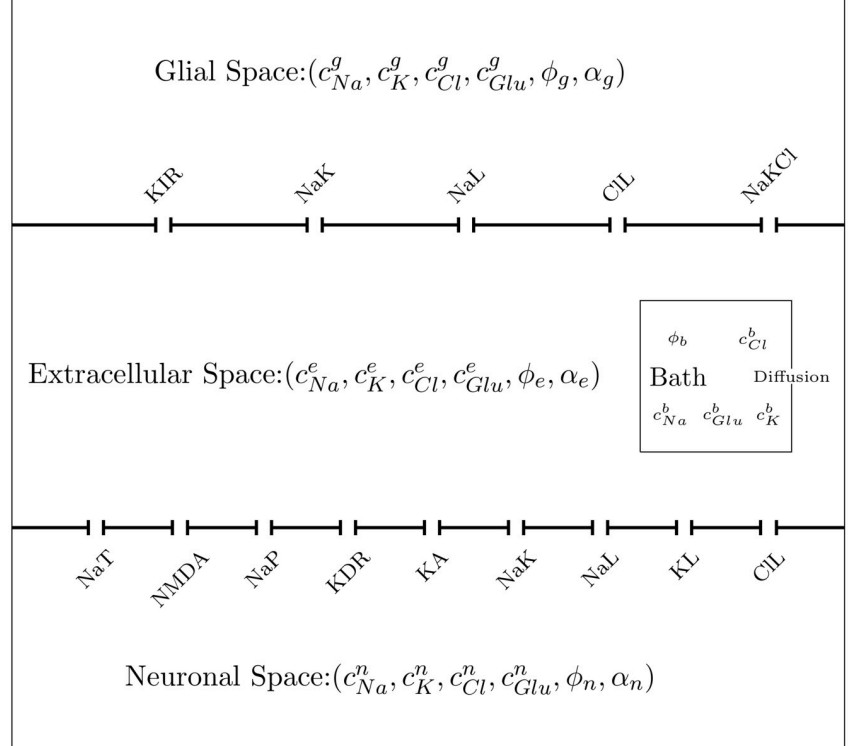

**Fig 1. A compartmental schematic of our model showing how the neurons and glia communicate with the extracellular space.**

Volume fractions change with transmembrane water flow:

$$\frac{\partial \alpha_k}{\partial t} = -\gamma_k w_k, \; k = \mathrm{n}, \mathrm{g}, \qquad \frac{\partial \alpha_\mathrm{e}}{\partial t} = \sum_{k=1}^{2} \gamma_k w_k. \tag{2}$$

Here, $\gamma_k$ represents the area of membrane per unit volume of tissue separating compartment $k$ from the extracellular space and $w_k$ is the water flow per unit area of membrane into each compartment $k$ from the extracellular space. The transmembrane water flux $w_k$ is proportional to the osmotic pressure difference between the extracellular space and compartment $k$:

$$w_k = \eta_k RT \left( \frac{a_\mathrm{e}}{\alpha_\mathrm{e}} + \sum_{i=1}^{M} c_i^\mathrm{e} - \frac{a_k}{\alpha_k} - \sum_{i=1}^{M} c_i^k \right), \; k = \mathrm{n}, \mathrm{g}$$

where $\eta_k$ is the hydraulic permeability and $a_k$ is the amount of immobile ions in compartment $k$.

Let $c_i^k$ be the concentration of the $i$th species of ion in the $k$th compartment (we use $Na^+$, $K^-$, $Cl^-$, and glutamate as our ions) and $\phi_k$ be the voltage in compartment $k$. For $i = 1, \cdots, M$, we have the following:

$$\frac{\partial (\alpha_k c_i^k)}{\partial t} = -\nabla \cdot f_i^k - \gamma_k g_i^k, \quad k = \mathrm{n}, \mathrm{g} \tag{3}$$

$$\frac{\partial (\alpha_\mathrm{e} c_i^\mathrm{e})}{\partial t} = -\nabla \cdot f_i^\mathrm{e} + \sum_{k=1}^{2} \gamma_k g_i^k - f_i^\mathrm{bath} \tag{4}$$

$$f_i^k = -D_i^k c_i^k \nabla \left( \ln c_i^k + \frac{z_i F}{RT} \phi_k \right), \quad k = \mathrm{n}, \mathrm{g}, \mathrm{e}, \tag{5}$$

$$f_i^\mathrm{bath} = -\frac{D_i^\mathrm{e}}{L_\mathrm{bath}^2} \left( \frac{c_i^\mathrm{e} + c_i^\mathrm{bath}}{2} \right) \left( \ln \left( \frac{c_i^\mathrm{e}}{c_i^\mathrm{bath}} \right) + \frac{z_i F}{RT} (\phi_\mathrm{e} - \phi_\mathrm{bath}) \right). \tag{6}$$

Here, $D_i^k$ is the diffusion coefficient and depends on the volume fraction $\alpha_k$, $RT$ is the ideal gas constant times temperature, $z_i$ is the valence of the $i$th ion and $F$ is the Faraday constant. The valence of $Na^+$, $K^+$ and $Cl^-$ are 1, 1, −1 respectively, and we set the valence of glutamate to be 0. The diffusion coefficient depends on the tortuosity and volume fraction as specified in S1 Text. Note that diffusion in the glial compartment models communication through gap junctions. Most regions of the central nervous system do not have extensive neuronal gap junctional coupling, and therefore, the neuronal diffusion coefficient is set to 0. We add an external bath that interacts with the extracellular through electrodiffusion, and serves as the ground we measure voltage against. The parameter $L_\mathrm{bath}$ can be interpreted as the distance to the bath. We take this value to be $L_\mathrm{bath} = 1\mathrm{cm}$. This value is somewhat arbitrary; larger values of $L_\mathrm{bath}$ do not appreciably change the features of the simulation. The $g_i^k$ represent the transmembrane ion fluxes due to ion channels, transporters, and ion pumps (for glutamate, the physiological meaning of $g_i^k$ is somewhat different, see below). These are functions of intracellular and extracellular ions and voltages and the individual models of these will be described later.

Next, we need an equation for the electrostatic potential. We use the following charge-capacitance equations:

$$\gamma_k C_m^k \phi_{ke} = z_0^k F a_k + \sum_{i=1}^{M} z_i F \alpha_k c_i^k, \quad \phi_{ke} = \phi_k - \phi_e, \quad k = n, g, \tag{7}$$

$$-\sum_{k=1}^{2} \gamma_k C_m^k \phi_{ke} = z_0^N F a_e + \sum_{i=1}^{M} z_i F \alpha_e c_i^e. \tag{8}$$

Here, $C_m^k$ is the membrane capacitance per unit area between the $k$th compartment and the extracellular space and $z_0^k$ is the average valence of the immobile charges.

## Ion channels

The transmembrane ion fluxes $g_i^k$ are a combination of fluxes from ion channels, transporters, and ion pumps. Ion channel currents in the neuron is given by:

$$g_{Na}^n = j_{NaL}^n + j_{NaP}^n + 2h_{NaK}^n + \frac{2}{3}j_{NMDA}^n$$

$$g_K^n = j_{KL}^n + j_{KDR}^n + j_{KA}^n - 3h_{NaK}^n + \frac{1}{3}j_{NMDA}^n$$

$$g_{Cl}^n = j_{ClL}^n$$

In Glia:

$$g_{Na}^g = j_{NaL}^g + 2h_{NaK}^g + j_{NaKCl}^g$$

$$g_K^g = j_{KIR}^g - 3h_{NaK}^g + j_{NaKCl}^g$$

$$g_{Cl}^g = j_{ClL}^g + 2j_{NaKCl}^g$$

Here, the $j_{NaL,KL,ClL}$ are leak channel flux, $j_{NaP}$ is the persistent sodium flux, $h_{NaK}$ are the active NaK ATPase pump flux, $j_{KA}$ is the transient potassium flux, $j_{KDR}$ is potassium delayed rectifier flux, $j_{NMDA}$ is the NMDA receptor flux, $j_{KIR}$ is the Potassium inward rectifier flux, and $j_{NaKCl}$ is the sodium-potassium-chloride cotransporter flux. The glutamate fluxes are discussed below in Section Glutamate dynamics. We note that the fast Na current is not included here. The fast Na current by itself inactivates too quickly and is not capable, in computational models, of producing sustained depolarizations that will generate SD. They are nonetheless important, especially in the interplay of SD with epilepsy for example, a topic that is beyond the scope of this study.

All of these fluxes have the form of:

$$j_i^k = m^p h^q P_{ion} J(c_i^k, c_i^e, \phi_{ke}), \; J = J_{HH} \text{ or } J_{GHK},$$

$$J_{HH} = RT \ln\left(\frac{c_i^k}{c_i^e}\right) + z_i F \phi_{ke}, \; J_{GHK} = \frac{c_i^k \exp(z_i F \phi_{ke}/RT) - c_i^e}{\exp(z_i F \phi_{ke}/RT) - 1}, \tag{9}$$

where $m$ and $h$ are gating variables, $P_{ion}$ is the permeability, $J$ is either the Hodgkin-Huxley (HH) or Goldman-Hodgkin-Katz (GHK) type of currents (they depend on inter/extra cellular concentration and the voltage difference). Each $m$ and $h$ has its own ODE that is of the form:

$$\frac{ds}{dt} = S(\phi_{ke}, s), \; s = m, h,$$

where $S$ is some linear function in $g$ with typical Hodgkin-Huxley type relations for opening

and closing of ion channels that depends on membrane voltage. A list of all parameters can be found in S1 Text.

## NMDA receptor and glutamate

Glutamate dynamics and NMDAR have long been known to play an important role in SD [15, 19], but modeling in this area has been sparse [16, 17, 26]. As is well-known, NMDAR is gated both by glutamate and voltage [27], and thus serves a coincidence detector thought to play an important role in memory formation [28]. NMDAR models often used in the simulation of SD treat glutamate gating in an indirect fashion, so that glutamate gating is directly influenced by extracellular potassium concentration [17] or neuronal membrane voltage [26]. To properly treat NMDAR glutamate gating, glutamate dynamics must be modeled. Here, we introduce a simple glutamate cycling model in which the glutamate released from neurons into the extracellular space is taken up by neurons and glia, and the glial glutamate is recycled back into the neurons.

**Glutamate dynamics.** Glutamate, after released by neurons into the extracellular space via synaptic release, is taken up by both neurons and glia. Glial glutamate is converted into glutamine and transported back to neurons via glutamine transporters on both the glial and neuronal membranes [29]. This glutamine is then converted back to glutamate inside the neurons. This is modeled by specifying the fluxes $g_{Glu}$ in Eq (3) as follows:

$$g_{Glu}^{e} = j_{Glu}^{syn} - j_{Glu}^{eg} - j_{Glu}^{en}, \tag{10}$$

$$g_{Glu}^{n} = -j_{Glu}^{syn} + j_{Glu}^{en} + j_{Glu}^{gn}, \tag{11}$$

$$g_{Glu}^{g} = j_{Glu}^{eg} - j_{Glu}^{gn}, \tag{12}$$

where

$$j_{Glu}^{syn} = A \frac{c_{Glu}^{n}}{c_{Glu}^{n} + \varepsilon} f_{syn}(\phi_{ne}), \tag{13}$$

$$j_{Glu}^{eg} = (1 - v)B_e(c_{Glu}^{e} - R_e c_{Glu}^{g}), \tag{14}$$

$$j_{Glu}^{en} = vB_e(c_{Glu}^{e} - R_n c_{Glu}^{n}), \ R_n = R_e R_g, \tag{15}$$

$$j_{Glu}^{gn} = B_g(c_{Glu}^{g} - R_g c_{Glu}^{n}). \tag{16}$$

We use the expression in [30] for the synaptic release flux $j_{Glu}^{syn}$. The constant $A$ controls the strength of synaptic release; this has been modified downward from [30] to achieve physiologically reasonable concentrations of extracellular glutamate. Following [30], we let the synaptic release of glutamate depend on neuronal membrane voltage as follows.

$$f_{syn}(\phi_{ne}) = (0.76mM)e^{-0.0044(\phi_{ne}-8.66)^2}. \tag{17}$$

The fluxes $j_{Glu}^{kl}$ are the glutamate recycling fluxes from compartment $k$ to compartment $l$. The flux $j_{Glu}^{gn}$ represents the glutamate flux from the glia to neurons via glutamine conversion; we have opted for a simple model that does not explicitly model glutamine concentration. The fluxes $j_{Glu}^{kl}$ are all proportional to $c_{Glu}^{k} - Rc_{Glu}^{l}$ for some constant $R$. The constant $R$ is not equal to 1, and reflect the fact that glutamate is actively partitioned into the different compartments, neuronal, glial, and extracellular. In the absence of synaptic flux $j_{Glu}^{syn}$, we see that $g_{Glu}^{k} = 0$ for all

$k$ if and only if:

$$c_{\text{Glu}}^{\text{g}} = R_{\text{g}} c_{\text{Glu}}^n, \quad c_{\text{Glu}}^{\text{e}} = R_{\text{e}} c_{\text{Glu}}^{\text{g}} = R_{\text{e}} R_{\text{g}} c_{\text{Glu}}^{\text{n}} = R_{\text{n}} c_{\text{Glu}}^{\text{n}}.$$

We note that setting $R_{\text{n}} = R_{\text{e}} R_{\text{g}}$ is the only choice that allows for a nontrivial solution to the above equations. The constants $R_{\text{e}}$ and $R_{\text{g}}$ are chosen to so that the steady state reflects the experimentally observed values of $10 \mu \text{mol}/\ell$ for $c_{\text{Glu}}^{\text{g}}$ [31], $0.01 \mu \text{mol}/\ell$ for $c_{\text{Glu}}^{\text{e}}$ [32] and $c_{\text{Glu}}^{\text{n}} = 10 \text{mmol}/\ell$ [32].

The rates $B_{\text{e}}$ and $B_{\text{g}}$ are the reabsorption rate and glutamate-glutamine cycle rate, and $v$ is a fraction of extracellular glutamate that get recycled back to the neurons near steady state. The value of $B_{\text{e}}$ is taken from [33]. For the glutamate-glutamine recycling rate, we have chosen to let $B_{\text{g}} = B_{\text{e}}/2$, given that glutamate in the glutamate-glutamine cycle must traverse two membranes, the glial and neuronal membranes. We have checked that the conclusions of our simulations is not sensitive to the choice of $B_{\text{g}}$ so long as it is greater than $B_{\text{e}}/2$ and is comparable in magnitude to $B_{\text{e}}$. The parameters and their values are summarized in Table 1. We note that the recent work of [16] includes a glutamate dynamics model, except that the authors focus on release of glutamate by glia through reverse uptake rather than neuronal release modeled here.

**NMDA receptor.** Our NMDAR model is based on [33], and we model NMDAR flux as follows:

$$j_{\text{NMDA}} = \hat{g}_{\text{NMDA}} \left( \frac{2}{3} J_{\text{Na}}^{\text{n,NMDA}} + \frac{1}{3} J_{K}^{\text{n,NMDA}} \right) \tag{18}$$

$$J_i^{\text{n,NMDA}} = P_{\text{NMDA}}^{\text{n}} \frac{F \phi_{\text{ne}}}{RT} \frac{c_i^{\text{n}} \exp\left( \frac{F \phi_{ne}}{RT} \right) - c_i^{e}}{\exp\left( \frac{F \phi_{ne}}{RT} \right) - 1}, \quad i = \text{Na}, \text{K} \tag{19}$$

$$\hat{g}_{\text{NMDA}} = G(\phi_{\text{ne}}) F_{\text{Glu}} y, \; F_{\text{Glu}} = \frac{\left( c_{\text{Glu}}^{e} \right)^{1.5}}{\left( c_{\text{Glu}}^{e} \right)^{1.5} + (2.3 \mu \text{mmol}/\text{l})^{1.5}}, \tag{20}$$

$$G(\phi) = \left( 1 + 0.28 \; \exp(-0.062\phi)([\text{Mg}^{2+}]_{\text{e}}/3.57\text{mmol}/\text{l}) \right)^{-1} \tag{21}$$

The function $G(\phi)$ encodes the voltage dependence of NMDAR gating (due to Mg block). The term $F_{\text{Glu}} y$ is the fraction of open channels, where the variable $y$ obeys the following

**Table 1. Glutamate-Glutamine cycle parameters.**

| Parameter | Description | Value |
|---|---|---|
| $v$ | Reabsorbtion Rate Percent | 0.1 [34] |
| $A$ | Release Rate | $50 mM/s$(adjusted from [30], see text) |
| $B_e$ | Decay Rate | $(42s)^{-1}$ [33] |
| $B_g$ | Cycle Rate | $(84s)^{-1}$ (see text) |
| $R_g$ | Glial Fraction | $10^{-3}$ (see text) |
| $R_e$ | Extracellular Fraction | $10^{-3}$ (see text) |
| $\varepsilon$ | Saturation Constant | $22.99 \mu M$ [35] [36] |
| $D_{\text{Glu}}$ | Glutamate Diffusion | $7.6 \times 10^{-6} \; cm^2/sec$ [37] |

differential equation.

$$\frac{dy}{dt} = k_2 D_1 - k_1 F_{\text{Glu}} y,$$

$$\frac{dD_1}{dt} = k_1 F_{\text{Glu}} y + k_4 D_2 - (k_2 + k_3) D_1,$$

$$\frac{dD_2}{dt} = k_3 D_1 - k_4 D_2.$$

The NMDAR transitions between the states $y$, $D_1$ and $D_2$. It is only state $y$ that contributes to current flow, $D_1$, $D_2$ are desensitized states. The fraction of open channels within state $y$ is given by $F_{\text{Glu}}$, and the rate of desensitization from state $y$ to state $D_1$ is proportional to this open fraction. Our assumption here is that the transition between the closed and open states within the state $y$ is sufficiently rapid [36]. The NMDA receptor parameters and their values are summarized in Table 2.

We also point out that the nature of the NMDAR desensitization may be due to Zn block [38].

We note that, in place of $F_{\text{Glu}} y$ in (20), [18] places a simple function of membrane voltage whereas [17] has a function of extracellular potassium. Such a choice does not allow for the study of the role of glutamate dynamics and NMDAR in SD generation and propagation.

## Simulation

**Discretization.** We simulate our equations via a mixed implicit-explicit finite volume routine, based on [13, 14]. We describe the numerical scheme in 2 spatial dimension. The 1D case is similar (and simpler). For a function $u$ defined on a Cartesian grid, let $u_{l,m}^n$ be the evaluation of our variables at position: $(x, y) = (l\Delta x, m\Delta x)$ and time $t = n\Delta t$. Define the discrete operators:

$$\left(\mathcal{D}_{\text{grad}}^+ u\right)_{l,m} = \left(\frac{u_{l+1,m} - u_{l,m}}{\Delta x}, \frac{u_{l,m+1} - u_{l,m}}{\Delta y}\right)$$

$$\left(\mathcal{D}_{\text{div}}^- v\right)_{l,m} = \frac{v_{l,m}^x - v_{l-1,m}^x}{\Delta x} + \frac{v_{l,m}^y - v_{l,m-1}^y}{\Delta y} \quad \text{where } v_{l,m} = (v_{l,m}^x, v_{l,m}^y),$$

$$\left(A^+ u\right)_{lm} = \frac{u_{l+1,m} + u_{l,m}}{2}.$$

**Table 2. NMDA receptor parameters.**

| Parameter | Description | Value |
|---|---|---|
| $P_{\text{NMDA}}$ | NMDAR Permeability | $0 - 6 \times 10^{-5}$ cm/sec |
| $[\text{Mg}^{2+}]_e$ | Magnesium Concentration | 2mM |
| $k_1$ | $y \rightarrow D_1$ | $3.94 s^{-1}$ |
| $k_2$ | $D_1 \rightarrow y$ | $1.94 s^{-1}$ |
| $k_3$ | $D_1 \rightarrow D_2$ | $0.0213 s^{-1}$ |
| $k_4$ | $D_2 \rightarrow D_1$ | $0.00277 s^{-1}$ |

The volume Eqs (1) and (2) are discretized as follows:

$$\alpha_k^{n+1} - \alpha_k^n + \Delta t w_k^{n+1} = 0, \quad n = 1, \ldots, N-1$$

$$\alpha_N^n = 1 - \sum_{k=1}^{N-1} \alpha_k^n \tag{22}$$

For the concentration Eqs (3) and (4), we have:

$$\alpha_k^{n+1} c_i^{k,n+1} - \alpha_k^n c_i^{kn} - \Delta t f_i^k + \Delta t g_i^{k,n+1} = 0, \quad k = \mathrm{n}, \mathrm{g}, \ i = 1, \ldots, M$$

$$\alpha_e^{n+1} c_i^{e,n+1} - \alpha_e^n c_i^{e,n} - \Delta t f_i^e - D_i^e \left( \frac{c_i^{e,n} + c_i^{\mathrm{bath}}}{2} \right) \left( \ln\left( \frac{c_i^{e,n+1}}{c_i^{\mathrm{bath}}} \right) + \frac{z_i F}{RT} \left( \phi_e^{n+1} - \phi_{bath} \right) \right)$$

$$-\Delta t \sum_{k=1}^{2} g_i^{k,n+1} = 0, i = 1, \ldots, M \tag{23}$$

$$f_i^k = \mathcal{D}_{\mathrm{div}}^- \left( D_i^k A^+ \left( c_i^{kn} \right) \mathcal{D}_{\mathrm{grad}}^+ \left( \ln\left( c_i^{k,n+1} \right) + \frac{z_i F}{RT} \phi_k^{n+1} \right) \right)$$

In the above, passive ionic currents in $g_i^K$ are treated implicitly in time whereas the active ionic currents are treated explicitly. Note that we write the flux $f_i^k$ in terms of the electrochemical gradient, and treat the electrochemical gradient implicitly, where as the coefficient, $D_i^k c_i^k$, is treated implicitly. This choice, taken in [13, 14], resulted in the most stable computations among the different discretization choices tested. The charge capacitance relations (7) and (8) are not discretized as is, but are discretized after taking the time derivative, and using (3) and (4) in the resulting expressions:

$$\gamma_k \frac{C_{\mathrm{m}}^k}{\Delta t} \left( \phi_{kN}^{n+1} - \phi_{kN}^n \right) - \sum_{i=1}^{M} z_i F \left( f_i^k - g_i^{k,n+1} \right) = 0, \quad k = \mathrm{n}, \mathrm{g}$$

$$-\frac{1}{\Delta t} \sum_{k=1}^{2} \gamma_k C_m^k \left( \phi_{ke}^{n+1} - \phi_{ke}^n \right) - \sum_{i=1}^{M} \sum_{k=1}^{2} z_i F g_i^{k,n+1} - \tag{24}$$

$$\sum_{i=1}^{M} z_i F \left( f_i^e + D_i^e \left( \frac{c_i^e + c_i^{\mathrm{bath}}}{2} \right) \left( \ln\left( \frac{c_i^e}{c_i^{\mathrm{bath}}} \right) + \frac{z_i F}{RT} \left( \phi_e - \phi_{\mathrm{bath}} \right) \right) \right) = 0$$

Eqs (23) and (24) are solved simultaneously as a nonlinear system of equations for the voltages $\phi_k$ and the concentrations $c_i^k$, which is solved using Newton's method with preconditioned Krylov subspace solvers (see below for detail). We note that the advantage of using the (24) in favor of a direct discretization of the algebraic relation (7) and (8) is that the capacitance $C_{\mathrm{m}}^k$ is very small in dimensionless terms [13]. Directly discretizing (7) and (8) results in an increasingly ill-conditioned system to be solved for $\phi_k$ and $c_i^k$ as $\Delta t$ is made small. Indeed, we have found that the above discretization significantly reduces the iteration count of the Krylov subspace solvers. Additionally, the gating variables depend on time. They are solved in a separate update step as:

$$s^{n+1} = s^n + \Delta t S(\phi_{ke}^{n+1}, s^{n+1})$$

Since $S$ is linear in $s$ we can directly solve the above equations. Fig 2 shows time profiles of an example 1D simulation. We can see the membrane voltage, the extracellular DC shift, the

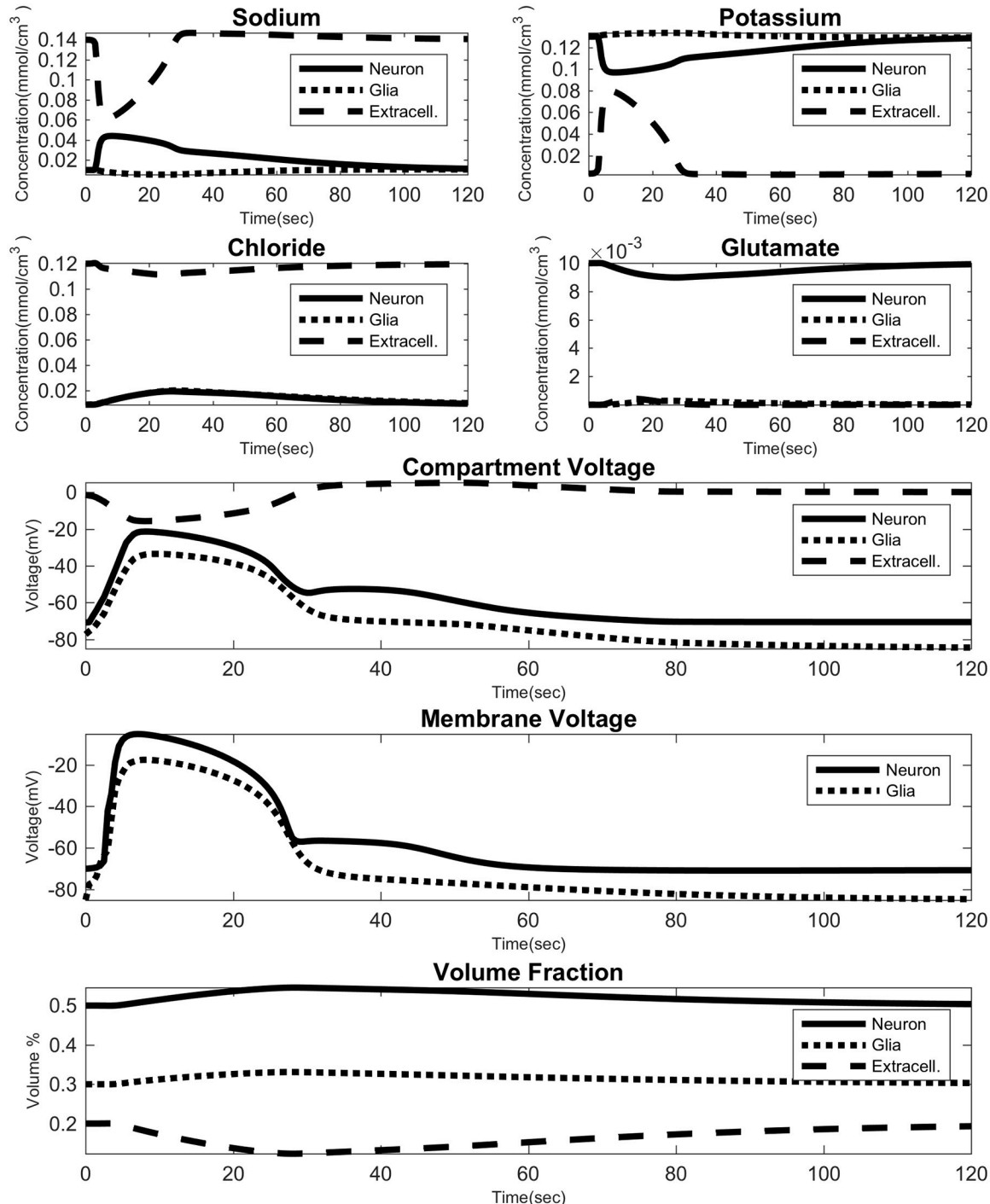

**Fig 2. Example 1D simulation.** The time course of all variables with $P_{NMDA} = 1 \times 10^{-5}$ cm/s and $P_{NaP} = 2 \times 10^{-5}$ cm/s. For a 0.5cm domain $\Delta x = 0.0156$ cm with $\Delta t = 0.01$ s. The time course is from the third grid point from the left end of the domain.

extracellular volume shrinkage, and the large ionic fluctuations. We initiate these waves by transiently increasing the membrane permeability of neurons to all ions on the left-most boundary of the domain for a short duration (0.5sec).

**Solvers.** As described above, each time step consists of the following substeps:

1. Update $\alpha_k$.

2. Update $c_i^k$ and $\phi_k$.

3. Update gating variables.

The updates of $\alpha_k$ and the gating variables do not involve any spatial coupling and thus consists simply of solving local nonlinear equations at each grid point.

On the other hand, the update of $c_i^k$ and $\phi_k$ requires the solution of a large spatially coupled nonlinear algebraic system, as discussed in the previous section. With three compartments (neuronal, glial, extracellular), 4 ions/molecules (Na$^+$, K$^+$, Cl−, glutamate) and the voltage as the unknowns, there are 15 unknowns at each grid point.

We use Newton's method to solve the nonlinear algebraic system. The Jacobian matrix in the Newton method is non-symmetric. We use preconditioned Krylov subspace solvers for the linear solvers [39]. The PETSc software package [25, 40, 41] is used, which provides a suite of powerful Krylov subspace solvers (KSP) and preconditoners. A full description of a specific solution routine involves describing: a. the method to solve the nonlinear equations (e.g Newton's Method), b. the linear solver method (e.g GMRES, LU), and c. the preconditioner used to make the iterative method converge faster (e.g incomplete LU, Multigrid). We have found the following setups lead to relatively fast performance on serial computer:

1. Grid sizes $2^k$, $k = 1, \cdots, 5$ or not power of 2.

    a. Nonlinear: Newton Line Search

    b. KSP: deflated GMRES

    c. Preconditioner: incomplete LU

2. Power of 2 bigger than 32.

    a. Nonlinear: Newton Line Search

    b. KSP: Flexible GMRES

    c. Preconditioner: W-Cycle Multigrid:

        i. SubKSP: Richardson

        ii. SubPreconditoner: SOR.

We have also performed convergence studies for certain test problems, which exhibited approximate 2nd order accuracy in space and 1st order accuracy in time. We refer the reader to [42] for details. The code can be found in the link provided in S3 Text.

## Results

### NMDAR and NaP in SD propagation

Here, we examine the relative importance of NMDAR and NaP in SD propagation. We first study the behavior of the velocity and duration of SD as we vary the expression level of NaP and NMDAR, with or without extracellular glutamate diffusion. In the presence of glutamate diffusion, we see that increased expression of both channels leads to increased velocity (Fig 3). NaP has a significantly greater impact on speed; purely NMDAR driven propagation has a speed of 20% that of purely NaP driven propagation [18]. The propagation speed when glutamate diffusion is set to zero is given in Fig 4. We can see that while the NMDAR has some

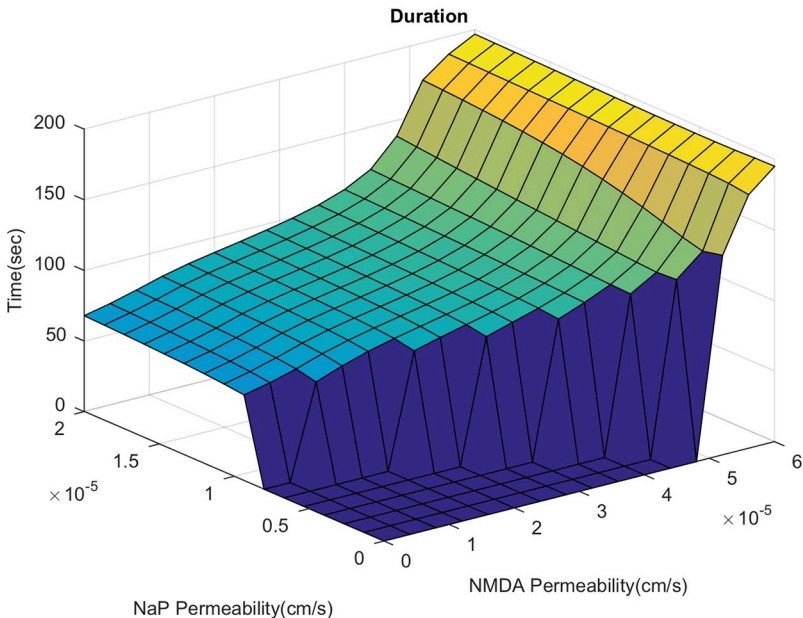

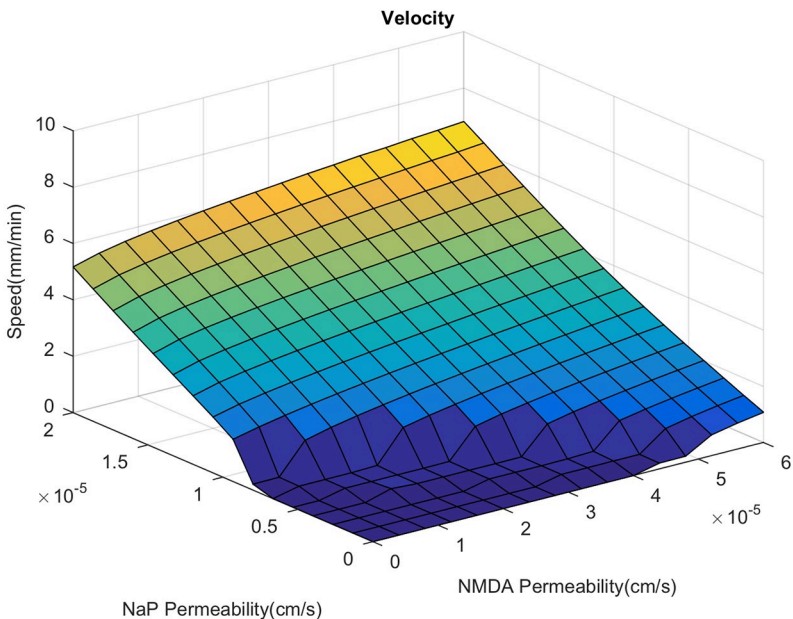

**Fig 3. Duration and velocity of spreading depression.** Varied over a range of $P_{NaP}$ and $P_{NMDA}$. Details on the calculation of duration and velocity are provided in S2 Text.

effect on propagation (most noticeably, allowing propagation to occur with slightly less NaP expression), its effect is significantly reduced.

The above results suggest that there are two modes of SD propagation in our model: NMDAR mediated propagation which is dependent on extracellular glutamate diffusion (see Fig 5), and NaP mediated propagation which works without glutamate diffusion and is primarily driven by ionic (potassium) diffusion (see Fig 6). Our results are consistent with the

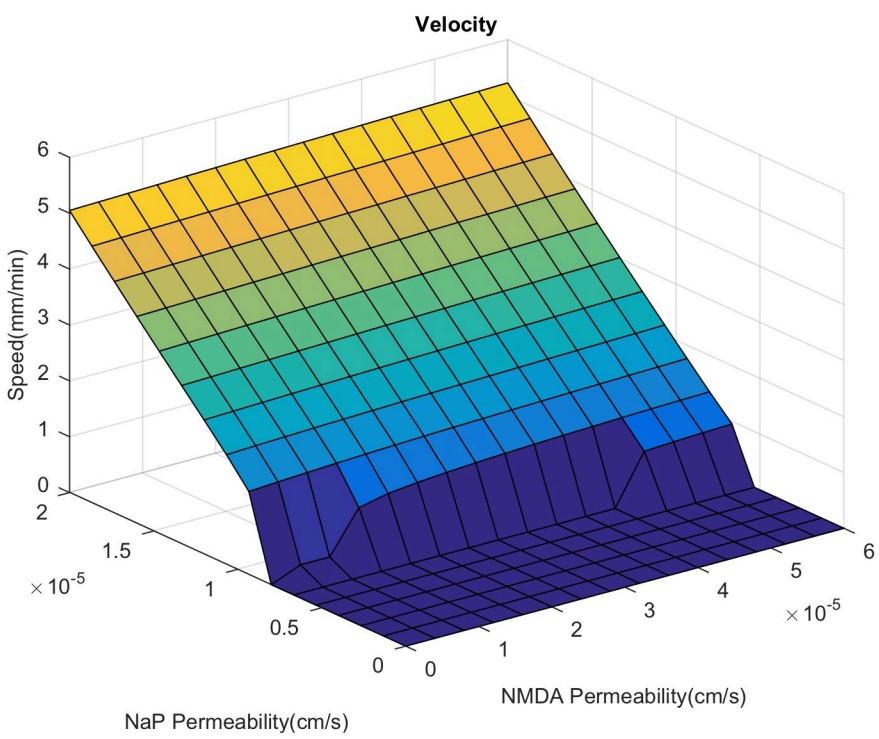

**Fig 4. Velocity of spreading depression without glutamate diffusion.** Varied over a range of $P_{\text{NaP}}$ and $P_{\text{NMDA}}$.

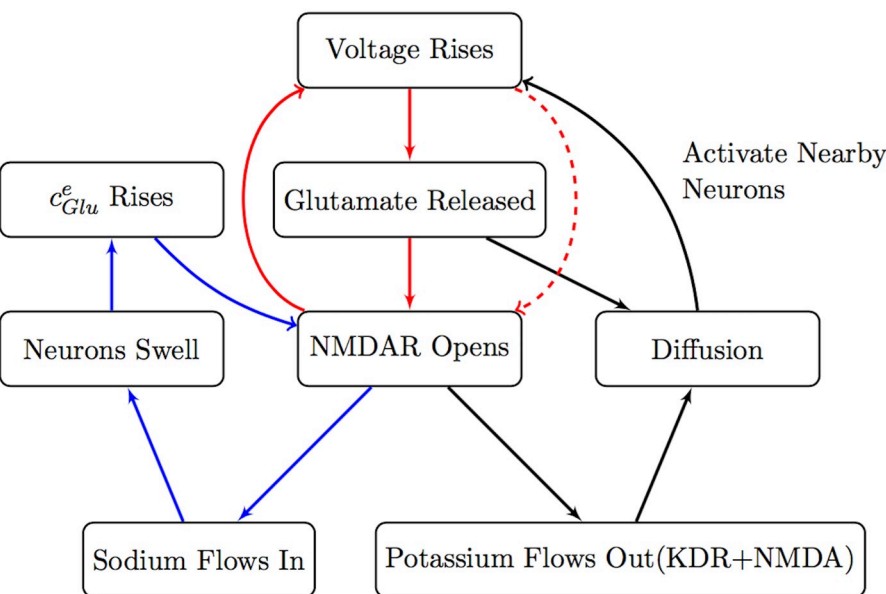

**Fig 5. NMDA receptor feedback mechanism.** Summary of CSD dynamics of our model. Initiation due to NMDA receptor with propagation caused by the combination of interstitial glutamate and potassium diffusion. Neuronal swelling causes prolonged activation of NMDA receptors.

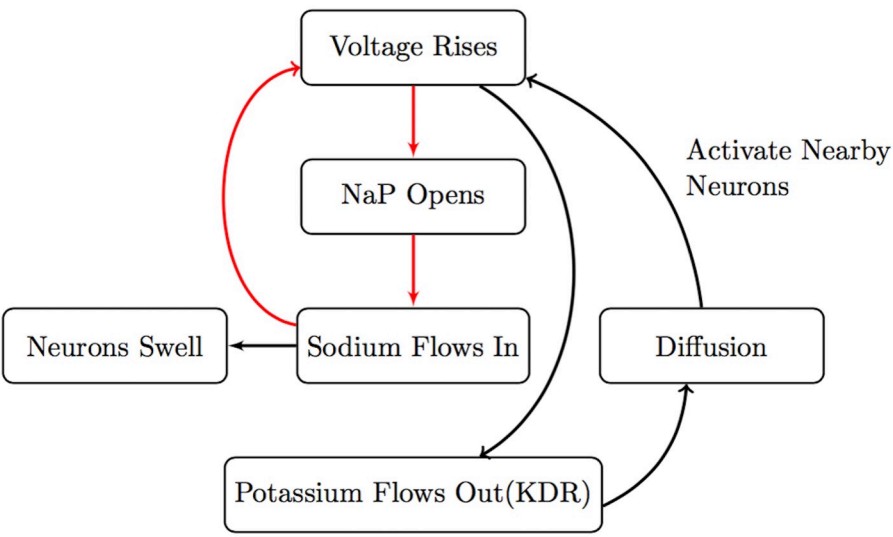

**Fig 6. Persistent sodium feedback scheme.** Summary of Initiation and Propagation due to the persistent sodium channel activation and interstitial potassium diffusion.

observation that normoxic and anoxic SD have different pharmacological properties [7]. Indeed, normoxic SD is suppressed by NMDAR antagonists whereas anoxic SD is suppressed by Na channel blockers. It is certainly the case, however, that the purely NMDAR mediated and the purely NaP mediated mechanisms we have identified here are two extremes, and that both mechanisms are at play to differing degrees in specific systems.

An important difference between the NMDAR and NaP mediated modes of propagation is the duration of the SD wave (Fig 3). As NMDAR increases we see an initially small increase in duration, and once we get to a certain level, the duration quickly increases from around 60 seconds to upwards of 180 seconds. This observations fits with experiments in the hippocampus [43], where the duration of SD in the dendrites (the stratum radiatum, in which NMDAR expression is high) is 2-3 times longer than the duration in the somata (the stratum pyramidale, in which NMDAR expression is low). As we shall see in the next section, this marked increase in duration corresponds to the appearance of a secondary delayed activation of NMDAR currents.

## SD time course and NMDAR expression

Increased NMDAR expression leads to a difference in the time course of the SD wave. As seen in Fig 7, the neuronal membrane voltage sees a second depolarization for high values of NMDAR expression. We shall henceforth focus on the time course of the extracellular DC shift, given its importance as an experimentally accessible observable. With NaP only, the SD wave has one valley. With higher NMDAR expression, a second delayed valley appears at around 60 to 100 seconds as seen in Fig 8. The initial DC shift (as well as the lone DC shift under purely NaP dynamics) reaches a minimum of $-15mV$, whereas the second shift goes below $-15mV$, nearing $-20mV$. We also see a positive overshoot in the recovery of the DC shift. Interestingly, this overshoot is most prominent for intermediate values of the NMDAR expression, and the overshoot is somewhat smaller for NMDAR expression levels that are higher (Fig 8).

The presence of two valleys in the extracellular DC shift ("inverted saddle") [6] is well-documented, but our result here seems to be the first to computationally reproduce this feature.

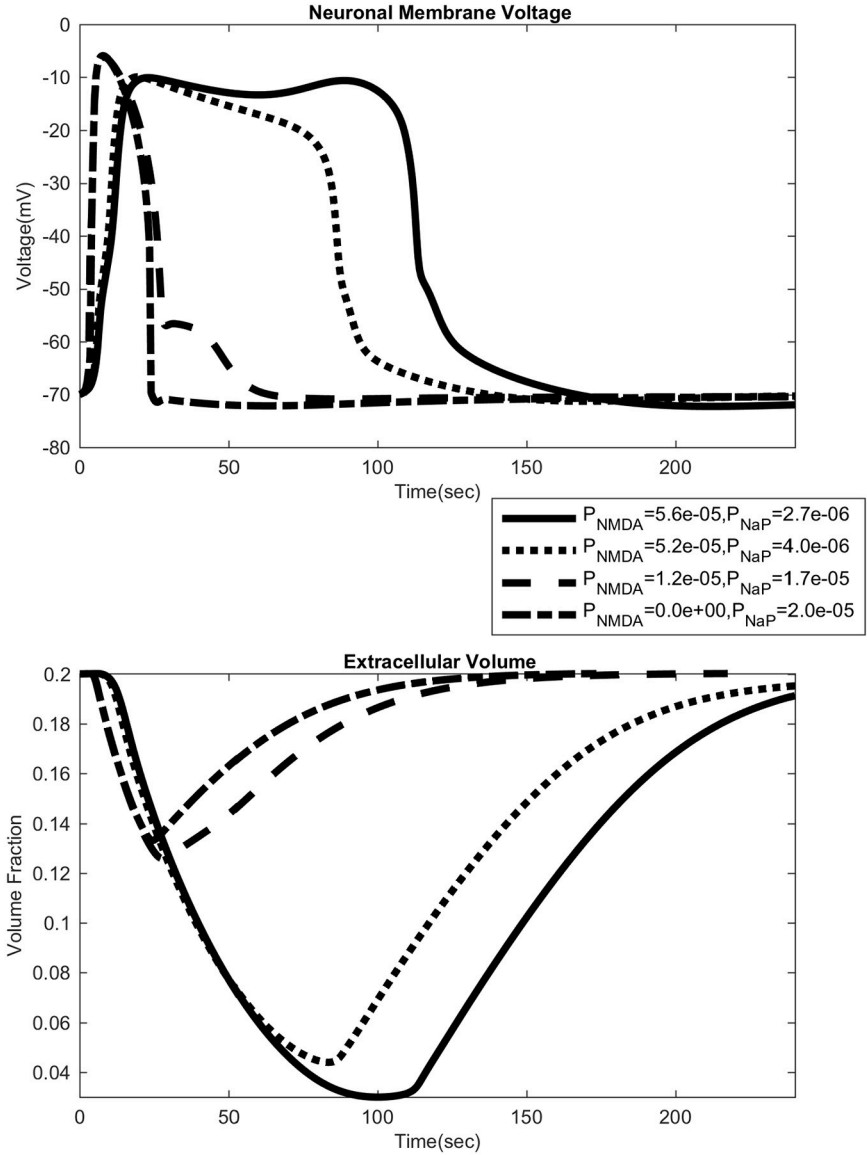

**Fig 7. Different time profiles of neuronal membrane voltage and extracellular volume.** Each panel shows time courses from four different levels of NMDAR and NaP. For neuronal membrane voltage, as NMDAR permeability increases a secondary bump appears. It appears for even smaller levels of NMDAR, barely visible on the dashed line. The volume graph shows the large reduction in extracellular space.

Our computational result indicates that the first peak is due mainly to NaP current whereas the latter is due mainly to NMDAR current. This is consistent with experiments showing that the DC shift differs depending on where the measurements are taken [6, 7, 44] or if the NMDA receptor is blocked [43]. Measurements taken near the dendrites, where NMDAR expression is higher, show this second valley, whereas measurements taken near the soma, where NMDAR expression is lower, do not show it. We also note that an overshoot in the DC shift is seen experimentally, and that its behavior with respect to NMDAR expression described herein is seen in [43] in their partially blocked NMDA-receptor experiment.

The SD time course and the presence of the second valley in the DC shift is strongly influenced by cellular volume changes. We varied the hydraulic permeability coefficient of the

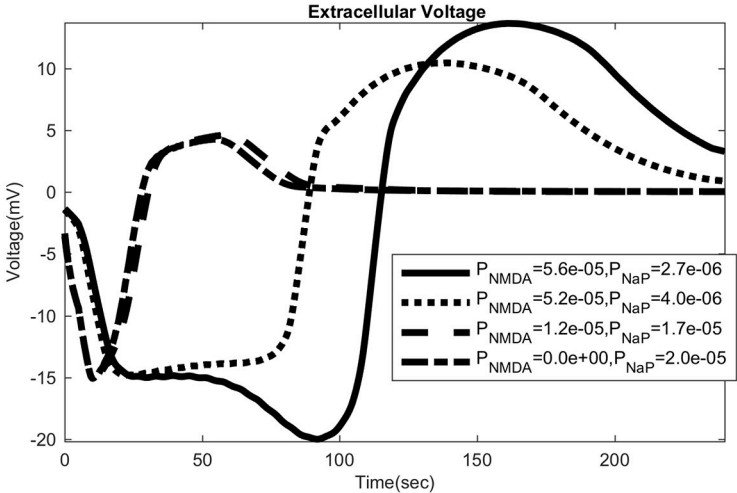

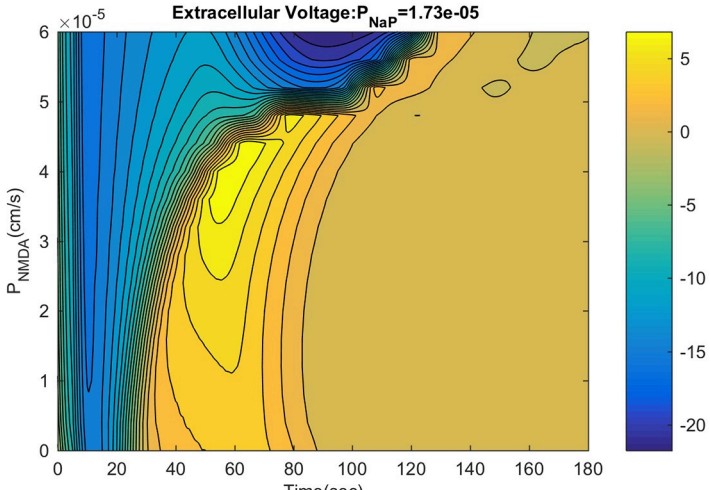

**Fig 8. Time profiles with extracellular voltage.** The figure on the top shows voltage traces for a sampling of NMDAR and NaP permeability. The figure on the bottom is the contour plot for the time course for extracellular voltage for different values of NMDAR permeability. Note that the overshoot is prominent for intermediate values of NMDAR permeability.

neuronal and glial membranes; a high permeability leads to greater volume changes. Without NMDAR, this has almost no effect besides reducing the expansion of neurons and glia. But, with NMDAR there is a prominent effect (see Fig 9). For large enough NMDAR permeability we see the two valleys in the DC shift, but as the hydraulic permeability is lowered, the second valley disappears. This may be explained as follows. NMDAR activation leads to cell swelling, which leads to shrinkage of the extracellular space, raising the glutamate concentration(Fig 10). Extracellular space constriction thus serves as a feedback loop which helps generate this secondary valley (see Fig 6).

## SD spiral

SD is a 3D phenomenon, and the 1D simulations conducted above and in previous studies cannot fully capture this phenomenon. Here, we consider the spatial patterns formed by SD

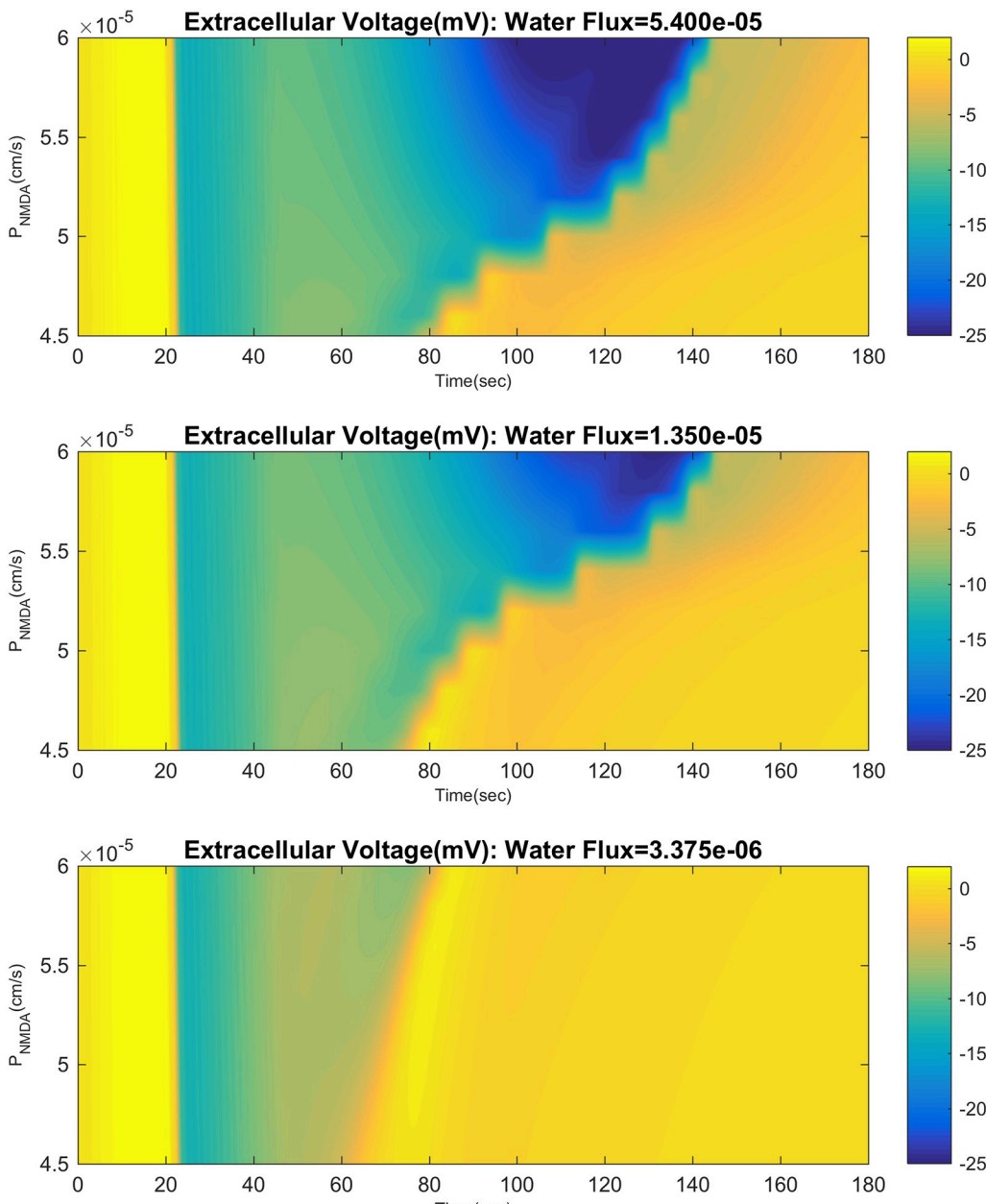

**Fig 9. Influence of cell swelling on extracellular voltage.** We vary NMDAR permeability between $4.5 - 6 \times 10^{-5}$ cm/s along the y-axis. Each panel has a different value for hydraulic permeability (water flux). The top and bottom panel have a minimum extracellular space of 2.5% and 10% respectively. For small enough hydraulic permeability, the wave looks no different than a NaP driven wave with no/little NMDA receptor activity.

waves in a 2D cross-section parallel to the cortical surface. 2D spiral and target patterns have been observed in experimental systems [4, 45] and the existence in vivo of such patterns are strongly suggested. Here, we focus on spiral patterns. Such patterns are interesting from a patho-physiological point of view in that, by definition, they never recover back to the spatially homogeneous rest state. SD spirals are thus likely to be detrimental to the affected neural tissue, as suggested by recent evidence on the correlation between repeated SD waves and poor prognosis after brain trauma [1, 2]. We also point out that spiral waves are intensively studies

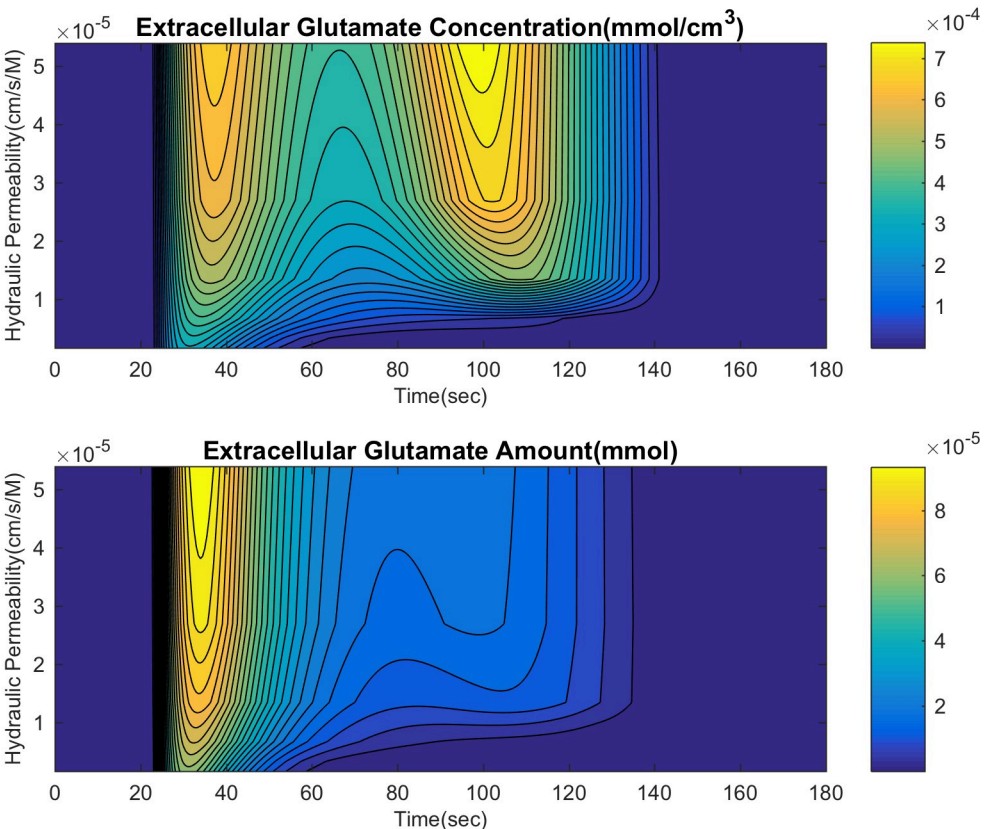

**Fig 10. Effect of varying hydraulic permeability on extracellular glutamate.** This shows the difference between the amount of glutamate(concentration times volume) and just concentration.

in cardiac electrophysiology [46]; the study of SD spirals may suggest interesting parallels between cardiac arrhythmias and SD.

To create a spiral, we first create an electrophysiologically refractory region in the center of the computational domain by transiently setting the inactivation gating variable of NaP and NMDAR permeability to 0. This prevents the SD wave from penetrating into this region. An SD wave is initiated at the lower half of the left side of the square computational domain, in the same way as the 1D wave. Once this region recovers, we are left with a self sustaining spiral. The behavior of the biophysical variables in a spiral is shown in Fig 11. It is not clear how spirals are formed in vivo, but the above is not an infeasible scenario. A block of tissue could become transiently inactive due to severe oxygen shortage, which could form the inactive core above. Indeed, it is well-known that spiral electrical activity in the heard is often triggered by ischemia or infarction.

To the best of our knowledge, this is the first computational demonstration of a spiral in a biophysically realistic SD model (see [4] for a computational study of SD spirals in a phenomenological model with nonlocal coupling). The computationally intensive nature of the spiral simulations limited the size of the domain to a.5 cm by.5 cm square with a simulation duration of up to 720 seconds in biophysical time. Some of the finer details of our simulation results, therefore, are not completely free of edge effects from the boundary of the computational domain or from the initialization of each computation. In the following, we thus focus on prominent overall trends that are insensitive to such details.

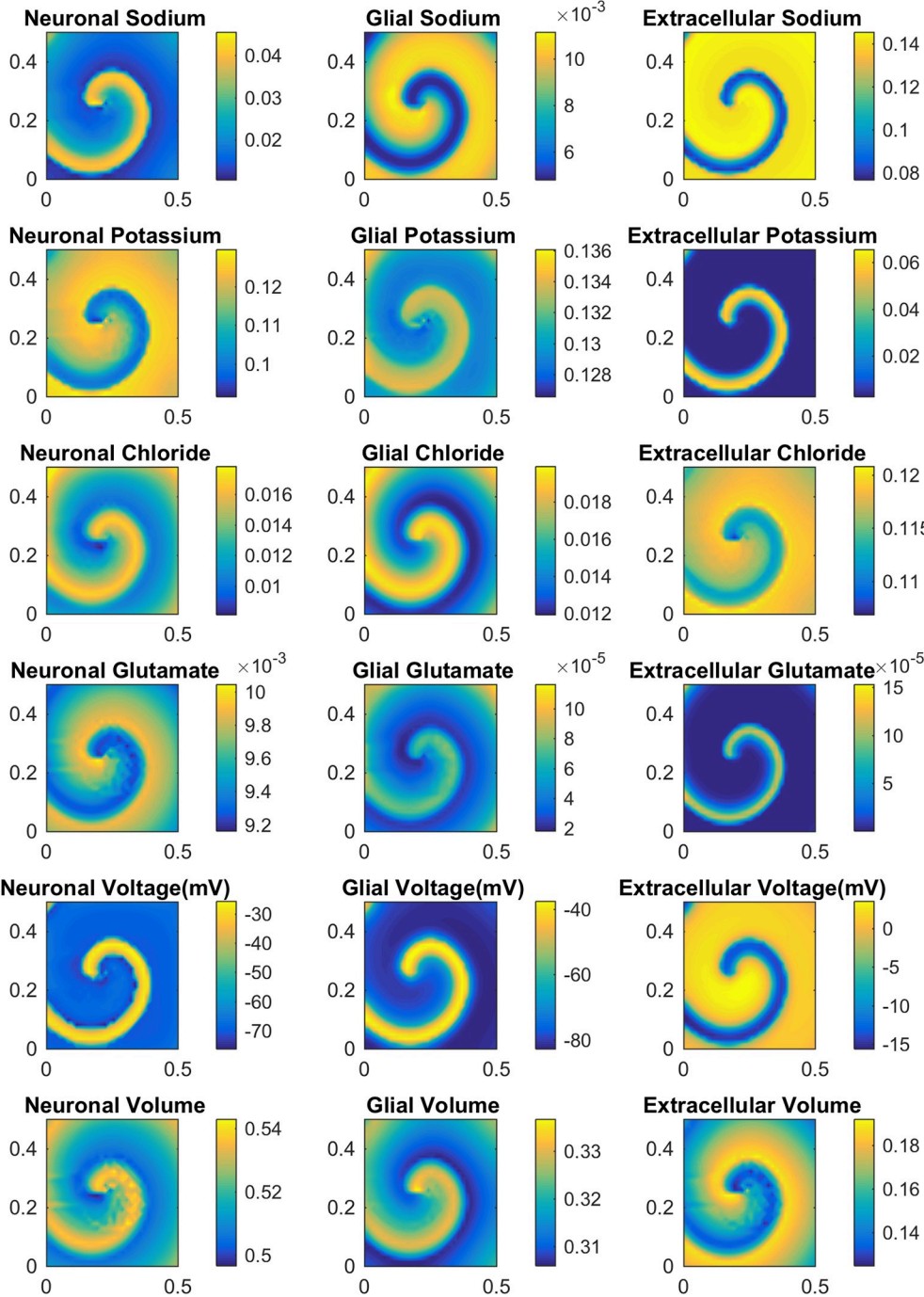

**Fig 11. Plot of major variables during a spiral.** All spiral simulations done with $\Delta x = \Delta y = 0.0156$cm and $\Delta t = 0.01$s.

## Speed of the SD spiral

We first compute the velocity of the spiral at each point in the computational domain. (for details on how we calculated the 2D velocity see S2 Text). Different points in the domain experience different velocities (Fig 12). The speed increases as we move away from the center, with the center moving at a speed near 0.5mm/min and the near-boundary moving no faster than

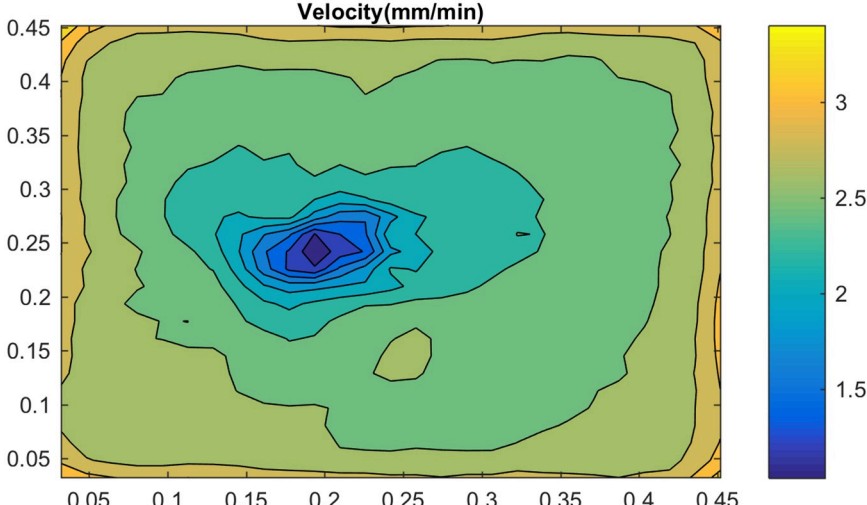

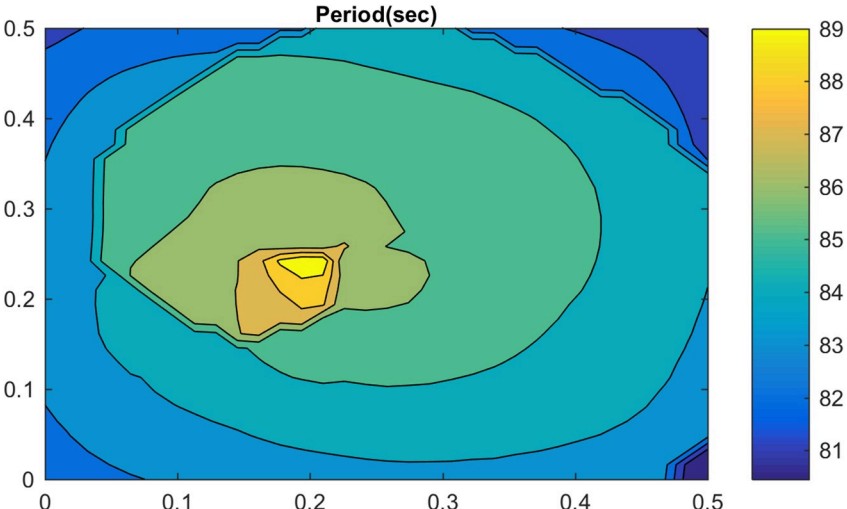

**Fig 12. Velocity and period of the wave at each point.** Speed of wave is calculated at each point in the domain (edges excluded due to edge effects, details provided in S2 Text). Period is calculated as the time between each depolarization for each point in the domain.

3.5mm/min. This maximum speed is a nearly 50% decrease in speed when compared to a plane wave with the same set of parameter values. We can also look at the period/frequency of excitation as well (Fig 12) and see that away from the core the time between excitations decreases.

The decrease in speed in comparison to the planar case is consistent with the experimental results in [45, 47] (chicken retina), where a loss in speed of 49% is reported. This decrease in speed is a consequence of the recurrent nature of the spiral; recurring excitations lead to slower waves since the tissue has not fully recovered from the last excitation.

We now investigate the change in spiral speed and duration as NaP and NMDAR is varied (Fig 13). The speed and duration are both lower as compared to the 1D case (compare with Fig 3). We also see that an increase in NaP does not lead to a large increase in speed as seen in the

**Average Velocity Over Domain**

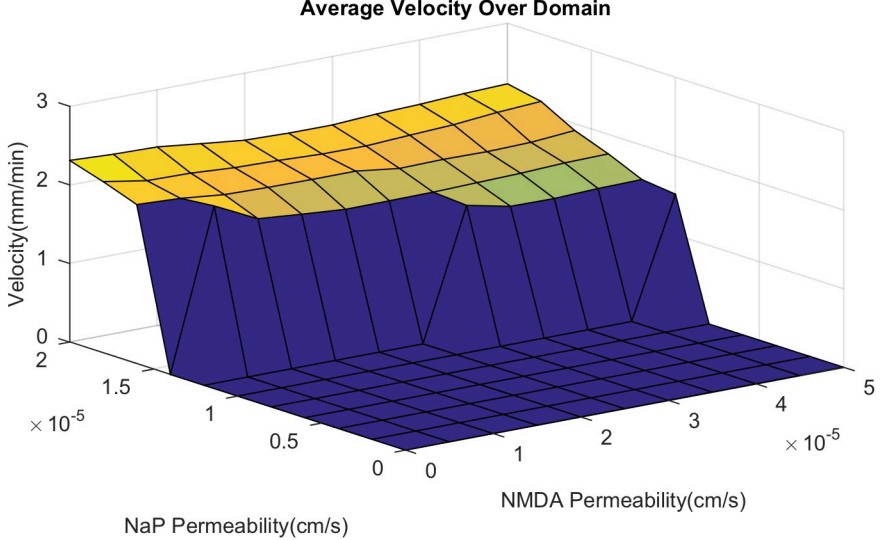

**Average Duration Over Domain**

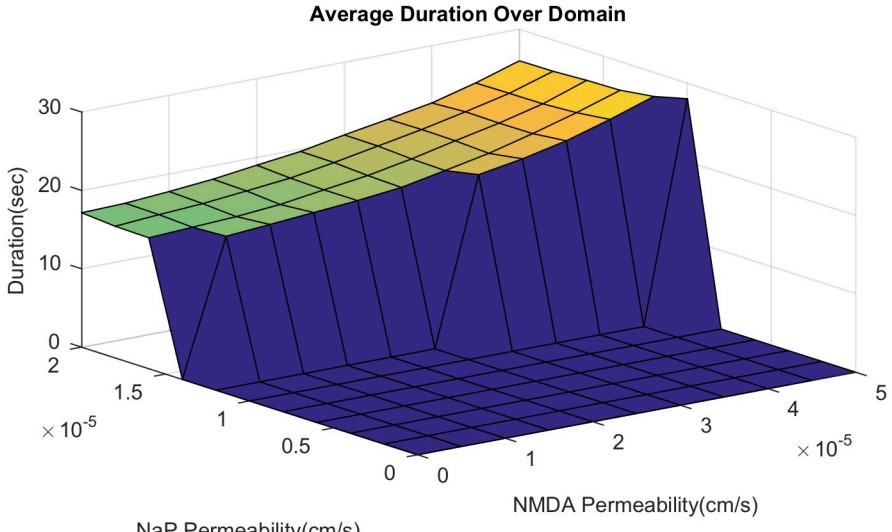

**Fig 13. Dependence of velocity and duration on NMDAR and NaP during a spiral.** Calculated by finding the average value over the whole domain. The zero sections are regions where the spiral dies off due to a lack of propagation. Beyond the NMDAR level shown in the above graphs, the duration becomes too long preventing the spiral from recurring.

1D case. This is a consequence of the fact that a faster wave implies that the next wave hits before the tissue has fully recovered, resulting in a slow down. In contrast to the 2D case, the change in duration with increased NMDAR is less dramatic; we lose the sharp increase in duration that we saw in the 1D case. We also note that the range of parameter values of NaP and NMDAR for which a spiral does not form is much larger than the corresponding range of propagation failure for 1D planar wave. Given the recurring nature of the SD spiral, a higher expression level of the active currents are needed for its generation. This may mean that only cortical areas that are highly susceptible to SD may experience SD spirals.

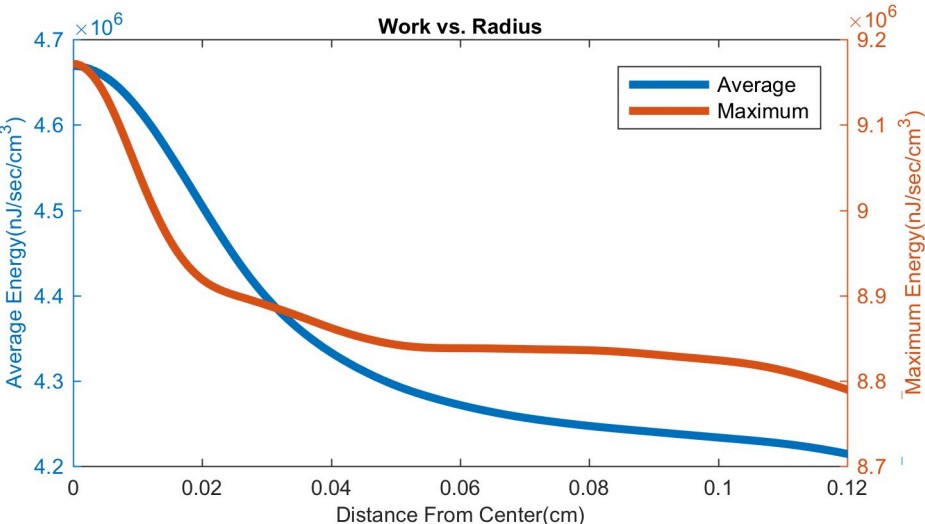

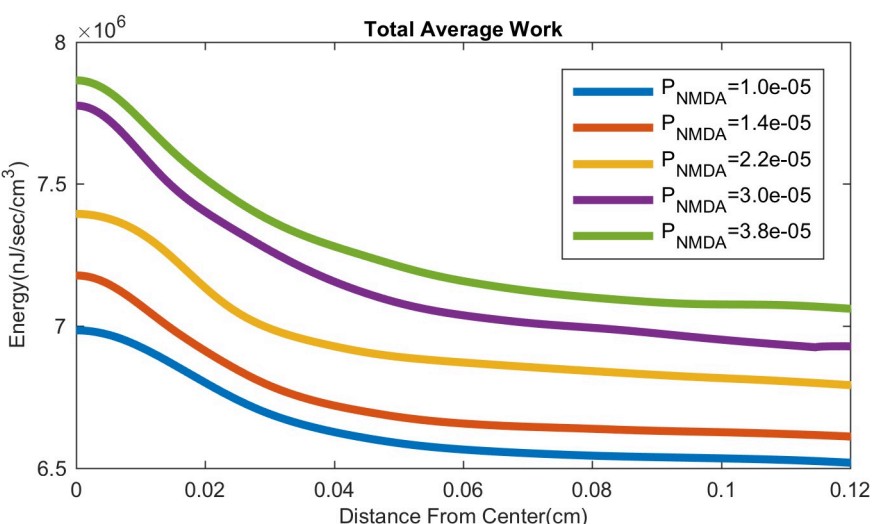

**Fig 14. Work done by ion pumps as a function of distance from center.** Top: Maximum and time average of the work done by ion pumps (note the two different *y*-axis scales), in both neurons and glia (over 3 minute duration), as measured by averaging over concentric circles around on the spiral center. Region closer to the center of the spiral does significantly more work. Bottom: Increasing NMDA receptor expression causes much more work to be done near the center, with a smaller increase seen away from the center.

## Energy consumption

Here, we compute the energy consumption due to ionic pumps as the spiral wave propagates through the computational domain. We note that this calculation is made possible by the fact that our model satisfies a free energy identity (see S2 Text and [13] for details). In Fig 14, we show the work done by the pumps as a function of distance from the center of the spiral. It is clearly seen that the work by the pumps is greater at the center than in the periphery. In Fig 14, we plot this radial profile as NMDAR expression is varied. The higher the NMDAR expression, the higher the overall energy consumption by the pumps.

SD and related phenomena have recently been identified as indicators or poor prognosis for patients suffering from stroke and traumatic brain injury [7]. The above computational

results suggests that recurrent spiral SD waves could particularly be damaging at the core of the spiral. The NMDAR study shows that NMDAR inhibitors can indeed have a neuroprotective effect, as reported in clinical trials [48].

## Discussion

In this paper, we introduced an electrodiffusion model of SD that includes glutamate and NMDAR dynamics, and performed 1D and 2D simulations. Our 1D simulations varying NaP and NMDAR expression in particular indicated that there are two modes of propagation, whose biophysical mechanism is summarized in Figs 6 and 5. NaP driven propagation relies primarily on extracellular $K^+$ diffusion whereas NMDAR driven propagation depends on glutamate diffusion and is strongly influenced by volume changes. NaP driven propagation is faster, and is of a shorter duration than the NMDAR driven propagation. These two propagation mechanisms are not completely separable and work in parallel. Indeed, the two valleys in the extracellular voltage signal seen in experiments can be explained by the presence of these two mechanisms; the first valley is primarily due to NaP activation and the latter due to NMDAR activation in our simulations. We point out that our prediction on the strong volume change dependence of NMDAR driven propagation could be tested experimentally, by studying the effect of extracellularly introduced impermeable sugars (mannitol etc.) on SD propagation. We then generated spirals and studied their properties in the 2D simulations as NaP and NMDAR are varied. The spirals are slower than their planar counterparts given their recurrent nature. We computed the energy consumption associated with the spirals, which indicate that the core experiences higher energy demand. We found that increased NMDAR expression leads to higher energy demand, which indicates the potential neuroprotective effect of NMDAR antagonists in recurring SD.

An important future direction is to improve the models of glutamate and NMDAR dynamics. A more biophysically faithful model will include detailed models of glutamate transporters as well as of the glutamine-glutamate conversion [16]. The greatest uncertainty in the model for NMDAR dynamics is in the treatment of long-term desensitization. One of the difficulties here is that SD is a phenomenon with a very long time course, whereas most experimental studies of NMDAR kinetics focus on the shorter time scale (of interest in normal electrophysiological conditions). Indeed, it must be pointed out that the mechanism of termination of SD is still unclear. In our model, NaP inactivation and NMDAR desensitization as well as glutamate cycling plays an important role, but this does not exclude the possibility that only a subset of these mechanisms, or even some different mechanism, may be responsible for recovery. In this connection, we also point out that the role of oxygenation and the vasculature is completely absent in our model [1, 20, 49, 50].

We have demonstrated that the computational framework we developed for the multidomain electrodiffusion model allows for biophysically detailed studies of SD. In the future, we will use our computational framework to investigate the impact of cortical layer structures on SD. The interplay of seizures and SD, studied at the level of an ordinary differential equation in [20], can be performed in the spatial setting by adapting our model. This would require the introduction of fast $Na^+$ currents, which we did not include in our model here. Fast $Na^+$ currents will require very fine time-stepping, which will lead to further challenges in the numerical method. Although the multidomain electrodiffusion model captures some of the major features of SD, it is important to ask what role microhistological features may play in SD; such microanatomy is likely to be even more important when considering the interplay between SD and seizures. Indeed, inclusion of fast Na currents at the coarse-grained level without microscale modeling implicitly assumes that all neurons residing in this mesoscopic area are

synchronized; this is likely not the case in many situations. Work on electrodiffusion modeling at the cellular level [51–53] can be of relevance in this regard.

## Supporting information

**S1 Text. Details of model.** Here, details of the ion channel models as well as the parameters used in the simulations are listed.
(PDF)

**S2 Text. Specifics of calculation.** A description of the calculation of velocity, duration and energy expenditure is given.
(PDF)

**S3 Text. Code.** Links to the simulation code are provided.
(PDF)

## Acknowledgments

The authors thank the IMA for hosting a workshop on SD in the February of 2018.

## Author Contributions

**Conceptualization:** Austin Tuttle, Jorge Riera Diaz, Yoichiro Mori.

**Formal analysis:** Austin Tuttle, Yoichiro Mori.

**Funding acquisition:** Jorge Riera Diaz, Yoichiro Mori.

**Investigation:** Austin Tuttle, Jorge Riera Diaz, Yoichiro Mori.

**Project administration:** Yoichiro Mori.

**Resources:** Yoichiro Mori.

**Software:** Austin Tuttle.

**Supervision:** Jorge Riera Diaz, Yoichiro Mori.

**Validation:** Austin Tuttle.

**Visualization:** Austin Tuttle.

**Writing – original draft:** Austin Tuttle, Yoichiro Mori.

**Writing – review & editing:** Austin Tuttle, Jorge Riera Diaz, Yoichiro Mori.

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
