## [Decision Letter · Decision Letter 0]

22 Jul 2019

Dear Dr Mori,

Thank you very much for submitting your manuscript 'A computational study on the role of glutamate and NMDA receptors on cortical spreading depression using a multidomain electrodiffusion model' for review by PLOS Computational Biology. Your manuscript has been fully evaluated by the PLOS Computational Biology editorial team and in this case also by independent peer reviewers. The reviewers appreciated the attention to an important problem, but raised some concerns about the manuscript as it currently stands. While your manuscript cannot be accepted in its present form, we are willing to consider a revised version in which the issues raised by the reviewers have been adequately addressed. We cannot, of course, promise publication at that time.

If you have any concerns or questions, please do not hesitate to contact us.

Sincerely,

Gaute T. Einevoll

Guest Editor

PLOS Computational Biology

Kim Blackwell

Deputy Editor

PLOS Computational Biology

[LINK]

Reviewer's Responses to Questions

**Comments to the Authors:**

Reviewer #1: This paper studied the role of glutamate and NMDA receptor dynamics in an electrodiffusion model of cortical spreading depression (SD) in both 1D and 2D spatial dimensions. They found that SD propagation depends on two distinct but overlapping mechanisms: 1) NaP driven SD propagation relies primarily on extracellular K+ diffusion; 2) NMDAR driven SD propagation depends on glutamate diffusion. For the first time, the “inverted saddle” signature of the extracellular voltage shift during SD can be explained by the coexistence of these two mechanisms: the first valley corresponds primarily to NaP activation and the latter valley to NMDAR activation. They also studied the properties of the spiral waves in 2D models. They found that the higher the NMDAR expression, the higher the overall energy consumption, which indicates the potential neuroprotective effect of NMDAR antagonists in recurring SD.

Overall, the article is well organized and presented. However, the following revisions should be considered:

1. About the description in Table 1 and 2, the authors should keep the first letter of second word either lowercase or uppercase consistently. The figure legends and subsection titles also have similar issues.

2. Page 9 from line 162 to 172, for the readers who are not using PETSc software package, it’s hard to understand these terms. It would be better if the authors explain a little bit more about them.

3. Page 10 from line 201 to 204, too many “where”, and it’s not quite easy to understand this sentence.

4. What is the advantage of using electrodiffusion model rather than Hodgkin-Huxley-type model?

5. About energy consumption, it is not clear how did the author gets the equation (24) and why dG/dt=-Ibulk-Imem. What is the difference of the energy consumption here and the ATP energy consumption due to the pump?

Reviewer #2: Review of the manuscript: «A computational study on the role of glutamate and NMDA receptors on cortical spreading depression using a multidomain electrodiffusion model».

The authors present a three-domain continuum model of neural tissue susceptible to spreading-depression (SD). The model includes a neuronal, glial and extracellular domain, and accounts for electrodiffusive ion transport within the domains, as well as for ion exchange between the domains via a set of ion channels, ion pumps and NMDA synapses. The simulations explore in a convincing way the differences and interplay between the propagation of the SD wave predicted by (1) persistent sodium activation + K+ diffusion, and (2) NMDA activation + Glutamate-diffusion, in a way which can explain various experimental observations. This is done both for a 1D planar wave and for 2D-spiralling waves (the latter of which this model is the first ever to simulate). A model like this should be welcomed by the neuroscience community as a tool for exploring and testing hypothesis regarding the mechanisms underlying the pathophysiology of SD.

The modelling work seems well conducted, and I would much like to see it published in PLoS CB. There are, however, some improvements that I think should be made to the manuscript beforehand, mainly in terms of how the model is introduced, how the modelling assumptions are discussed, and how the simulations are related to the biological system/phenomenon at hand. I think this would amount to a minor to moderate revision.

####### Moderately Major points:

##### 1 Relating to biology

1.1. I miss in the introduction a brief overview of the neurophysiology of SD. One thing that confuses me a bit is the role of Nap in this and previous models of the authors, and also the previous (cited) model by Kager et al. This may reflect my lack of knowledge, but I thought that the common view of SD in relation to the K+ diffusion hypothesis was that the membrane depolarization came from AP-firing (e.g., via Nat and Kdr) and the following increase in the K+ reversal potential due to a gradual increase in extracellular K+ (see e.g., Ayata & Lauritzen 2015). There are probably good reasons to exclude APs in a coarse-grained model like this. However, should Nap be understood as a playing the role as a stand-in mechanism for Nat, e.g. does Nap sort of work as a temporally averaged version of Nat on a long time scale, or is it in itself the key mechanism? That is, is the depolarization (in biology and in the model) explained mainly by a persistent sodium current or by K+ reversal potential changes which could follow from AP firing?

1.2. I will not demand it, but I think a schematic figure 1 that illustrates what goes into the model would help the reader to get into the material.

1.3. Fig. 1 is introduced with (line 143-144): “Since S is linear in s we can directly solve the above equations. Fig. 1 shows time profiles of an example 1D simulation”. I think the simulation should be explained more carefully to help the reader get into the material. How was SD triggered in the system? Was there an input signal? What was the recording position?

1.4. Likewise, the spiral simulations (line 242-243) were introduced with: “In two spatial dimensions, we can obtain more interesting dynamics. In the following, we focus on spirals. To create a spiral, we first create an electrophysiologically refractory region in the center of the computational domain.” The way this is written makes it appear like the spirals were created “just for fun”. I would suggest that this subsection is introduced in a more biology-oriented fashion, i.e. by briefly explaining that such spirals have been observed in biological systems, and as such motivating why one would like to simulate them.

1.5. Relating to the above quite, it is unclear to me how “an electrophysiological refractory region” was created in the center of the computational domain, and what it represents biologically. This must be more carefully explained.

1.6. Line 279-281 read: “We also note that the range of parameter values of Nap and NMDAR for which a spiral does not form is much larger than the corresponding range of propagation failure for 1D planar wave. Given the recurring nature of the SD spiral, a higher expression level of the active currents are needed for its generation.” This is a nice model prediction. Can it be tied to some experimental observations, or discussed in terms of where/when we see spirals or planar waves?

1.7. Generally (like in my points 1.3-1.5), the simulated results could be described in some more detail, and could be related more strongly to the biological scenario that is being simulated. Such improvements are likely to increase the impact that this work can have on the community.

1.8. In Fig. 13, for small distances, the Average work is larger than the Maximum work. This seems wrong. Is it something here that I don´t understand?

#### 2 Eletrodiffusive continuum model

By necessity, a coarse grained model like this rests on a series of assumptions regarding how the detailed structures and morphologies of neuropil can be collapsed into a tree-domain continuum in a meaningful manner. Whereas I believe that most of the assumptions made in the proposed model are sound and motivated, I feel like several of them could be stated and discussed in some further detail.

2.1. As suggested by eq. 5 continuous electrodiffusion occurs internally in all three domains. While it has previously been motivated that transport through the extracellular space and through the gap-junction coupled astrocytic syncytium can be represented as a continuums (Chen & Nicholson 2000), I have not seen a corresponding motivation for such continuous transport in neurons. I think this at least should be commented on when introduced, and perhaps given a brief discussion.

2.2. The tortouosity given in Table 4 suggests that the porous medium approximation was used, but the tortouosity does not appear in eq. 5. Was or wasn´t it part of the model?

2.3. Eq. 4 suggests that the extracellular space interacted with an omnipresent bath solution. This assumption appears somewhat bold, and should be discussed when introduced. Is it a technicality, or does the magnitude of the bath interaction play an important role for the simulation outcome? Was it tuned? What could it represent? Blood vessels, which are not part of the model, or leakage to deeper layers or into white matter?

2.4. The code for the model should be made available online. I think it has the potential to be a much used tool for exploring SD by many labs, at least if it is moderately easy to download and run the code.

#### 3. Language

Some parts of the manuscript seems hastily written, and the authors should do a revision before the final version is submitted. Especially the figure texts are often very sparse, sometimes should provide more information.

#### 4. Minor comments

Fig 6: “Each graph is 4 different NMDAR” should read: “Each panel shows four different NMDAR”.

Fig 10: “have very wide arcs that the main drivers do not have”: I was not able to understand this from looking at the figure. I think the arcs should be explained more carefully in the main text, since these results are not easy to wrap ones head around.

Fig 11: “Period of each point, time between each depolarization for each

point in the domain.” This should be made into a sentence and related explicitly to something that we see in the figure.

Fig. 13: “by averaging around the spiral center”. Should this be averaging over a circle around the spiral center? Also: “with a smaller increasing” should be “with a smaller increase”.

Line 46: “in either 1D or 0D”. Whereas I understand what 0D means, it seemed a bit odd. Perhaps explain that this is about point models with no spatial extension?

Eq. 3-5. Is it necessary to use two notations (1,2,3 or n,g,e) for index k? It would be tidier to use n,g,e consistently.

Line 116-117: Typo: “Bg greater than Bg.”

Line 198: “An important difference between the two modes of propagation”. I think you should define here what the two modes are.

Line 303: “NMDAR driven propagation depends on glutamate diffusion and is strongly influenced by volume changes.” Is this not the case for K+ diffusion? Please comment!

Line 337: “Fast Na+ currents will require very fine time-stepping, which will lead to further challenges in the numerical method.” Again, I wonder here if Nap could essentially have the same long term effect as a (temporally smeared) seizure. If appropriate, this could be discussed here. Also, I suppose that the inclusion of fast Na-currents in a continuum-model is also poses some conceptual problems? For example, in 1D, does it not effectively correspond to the assumption that all neurons (at a given x) fire completely synchronous APs?

Ayata, C., & Lauritzen, M. (2015). Spreading Depression, Spreading Depolarizations, and the Cerebral Vasculature. Physiological Reviews, 95(3), 953–993. https://doi.org/10.1152/physrev.00027.2014

Chen, K. C., & Nicholson, C. (2000). Spatial buffering of potassium ions in brain extracellular space. Biophysical Journal, 78(6), 2776–2797. https://doi.org/10.1016/S0006-3495(00)76822-6

**Have all data underlying the figures and results presented in the manuscript been provided?**

Reviewer #1: Yes

Reviewer #2: No: Model code should be made available through an online depository

PLOS authors have the option to publish the peer review history of their article (what does this mean?). If published, this will include your full peer review and any attached files.

Reviewer #1: No

Reviewer #2: Yes: Geir Halnes

---

## [Decision Letter · Decision Letter 1]

2 Oct 2019

Dear Dr Mori,

We are pleased to inform you that your manuscript 'A computational study on the role of glutamate and NMDA receptors on cortical spreading depression using a multidomain electrodiffusion model' has been provisionally accepted for publication in PLOS Computational Biology.

In the meantime, please log into Editorial Manager at https://www.editorialmanager.com/pcompbiol/, click the "Update My Information" link at the top of the page, and update your user information to ensure an efficient production and billing process.

One of the goals of PLOS is to make science accessible to educators and the public. PLOS staff issue occasional press releases and make early versions of PLOS Computational Biology articles available to science writers and journalists. PLOS staff also collaborate with Communication and Public Information Offices and would be happy to work with the relevant people at your institution or funding agency. If your institution or funding agency is interested in promoting your findings, please ask them to coordinate their releases with PLOS (contact ploscompbiol@plos.org).

Thank you again for supporting Open Access publishing. We look forward to publishing your paper in PLOS Computational Biology.

Sincerely,

Gaute T. Einevoll

Guest Editor

PLOS Computational Biology

Kim Blackwell

Deputy Editor

PLOS Computational Biology

Reviewer's Responses to Questions

**Comments to the Authors:**

Reviewer #1: The authors significantly improved the manuscript and addressed each of my comments. I only have one more minor issue, the figure resolution is too low and figure size is too large. It looks like it is larger than A4 size. The authors should address the issue before publication.

Reviewer #2: The authors have appropriately addressed all the concerns that I had with the original submission, and I suggest that the paper is accepted for publication in PLoS CB.

**Have all data underlying the figures and results presented in the manuscript been provided?**

Reviewer #1: None

Reviewer #2: Yes

PLOS authors have the option to publish the peer review history of their article (what does this mean?). If published, this will include your full peer review and any attached files.

Reviewer #1: No

Reviewer #2: Yes: Geir Halnes

---

## [Editor Report · Acceptance letter]

25 Nov 2019

PCOMPBIOL-D-19-00996R1 

A computational study on the role of glutamate and NMDA receptors on cortical spreading depression using a multidomain electrodiffusion model

Dear Dr Mori,

I am pleased to inform you that your manuscript has been formally accepted for publication in PLOS Computational Biology. Your manuscript is now with our production department and you will be notified of the publication date in due course.

With kind regards,

Matt Lyles
